# Surface Kondo effect and non-trivial metallic state of the Kondo insulator $YbB_{12}$

Kenta Hagiwara[1], Yoshiyuki Ohtsubo[1,2], Masaharu Matsunami[3], Shin-ichiro Ideta[3], Kiyohisa Tanaka[3], Hidetoshi Miyazaki[3], Julien E. Rault[4], Patrick Le Fèvre[4], François Bertran[4], Amina Taleb-Ibrahimi[4,5], Ryu Yukawa[6], Masaki Kobayashi[6], Koji Horiba[6], Hiroshi Kumigashira[6], Kazuki Sumida[7], Taichi Okuda[8], Fumitoshi Iga[9] & Shin-ichi Kimura[1,2]

A synergistic effect between strong electron correlation and spin–orbit interaction has been theoretically predicted to realize new topological states of quantum matter on Kondo insulators (KIs), so-called topological Kondo insulators (TKIs). One TKI candidate has been experimentally observed on the KI $SmB_6(001)$, and the origin of the surface states (SS) and the topological order of $SmB_6$ has been actively discussed. Here, we show a metallic SS on the clean surface of another TKI candidate $YbB_{12}(001)$ using angle-resolved photoelectron spectroscopy. The SS shows temperature-dependent reconstruction corresponding to the Kondo effect observed for bulk states. Despite the low-temperature insulating bulk, the reconstructed SS with $c–f$ hybridization is metallic, forming a closed Fermi contour surrounding $\bar{\Gamma}$ on the surface Brillouin zone and agreeing with the theoretically expected behaviour for SS on TKIs. These results demonstrate the temperature-dependent holistic reconstruction of two-dimensional states localized on KIs surface driven by the Kondo effect.

[1] Department of Physics, Graduate School of Science, Osaka University, Toyonaka 560-0043, Japan. [2] Graduate School of Frontier Biosciences, Osaka University, Suita 565-0871, Japan. [3] UVSOR Facility, Institute for Molecular Science, Okazaki 444-8585, Japan. [4] Synchrotron SOLEIL, Saint-Aubin-BP 48, F-91192 Gif sur Yvette, France. [5] UR1/CNRS-Synchrotron SOLEIL, Saint-Aubin, F-91192 Gif sur Yvette, France. [6] Photon Factory, Institute of Materials Structure Science, High Energy Accelerator Research Organization (KEK), 1-1 Oho, Tsukuba 305-0801, Japan. [7] Graduate School of Science, Hiroshima University, 1-3-1 Kagamiyama, Higashi-Hiroshima 739-8526, Japan. [8] HiSOR, Hiroshima University, Higashi-Hiroshima 739-0046, Japan. [9] College of Science, Ibaraki University, Mito 310-8512, Japan. Correspondence and requests for materials should be addressed to Y.O. (email: y_oh@fbs.osaka-u.ac.jp) or to S.-i.K. (email: kimura@fbs.osaka-u.ac.jp).

Recently, non-trivial surface electronic structures such as surface metallic states of topological insulators[1,2] and giant Rashba-type spin splitting of polar semiconductors[3–5] have been observed. These electronic structures originate from the large spin–orbit interaction (SOI) of heavy elements. On the other hand, in metallic compounds containing heavy elements, especially rare-earths, heavy quasiparticles, namely heavy fermions, appear owing to the Kondo effect[6]. The origin of the Kondo effect is a hybridization between the conduction band and localized states (for example, $4f$ states of rare-earths), namely, $c$–$f$ hybridization, derived from the Anderson model. At the surface of rare-earth intermetallic compounds, new physical properties originating from the Kondo effect and the large SOI such as superconductivity without inversion symmetry[7] will appear. Kondo insulators (KIs) are one candidate of such compounds.

In general, KIs possess a small energy gap (typically a few tens of meV) at the Fermi level ($E_F$) owing to $c$–$f$ hybridization at low temperatures[8]. In the case that the $c$–$f$ gap is formed by the conduction and valence bands with inverted parities, two-dimensional (2D) metallic electronic states should always appear on the KI surface as in the case of topological insulators[9,10]. Such materials are categorized as topological Kondo insulators (TKIs) in which the topological surface states (SS) originate from $c$–$f$ gap formation owing to the Kondo effect. Thus, TKIs are new physical states of quantum matter driven by the synergistic effect between strong electron correlation and SOI.

The surface state of the KI samarium hexaboride ($SmB_6$) has been investigated both theoretically and experimentally[11–16] and has been theoretically predicted to be a non-trivial topological surface state of a TKI. This predicted surface electronic structure has been experimentally observed[12–15]. However, the origin of the metallic surface state is currently under debate because Hlawenka *et al.* recently reported that the surface metallic state trivially originates from large Rashba splitting[16]. Therefore, a survey of another material is desirable to provide further insight into the origin of metallic SS on KIs.

Ytterbium dodecaboride ($YbB_{12}$) is a typical KI which has a NaCl-type crystal structure with Yb and $B_{12}$ clusters[17]. A clear energy gap appears in the bulk of $YbB_{12}$ with a gap size of about 40 meV of the peak (15 meV of the onset), which has been observed by an optical conductivity measurement[18]. Using angle-integrated photoemission measurements, a pseudo-gap opens at $E_F$, but a finite density of states has been observed at temperatures lower than that of the full gap opening[19]. The observed finite density of states at $E_F$ is considered to originate from a metallic surface state, and this metallic surface state has been confirmed using electrical transport measurements (Iga—private communication). Moreover, a theoretical study has predicted that this metallic surface conductivity originates from topological SS[20]. However, the band structure of $YbB_{12}$ has not been observed using momentum-resolved measurements such as angle-resolved photoelectron spectroscopy (ARPES) because a well-defined clean surface has not been obtained (ARPES spectra at only two $k$-points have been reported in ref. 21).

In this article, we report a surface state (SS) on a clean surface of the KI $YbB_{12}$(001) and its temperature-dependent reconstruction based on ARPES. We succeed in obtaining a clean $YbB_{12}$ surface which is suitable for detailed ARPES study. The SS is metallic and showed no dispersion along $k_z$, indicating its surface localization. While the SS does not hybridize with the Yb $4f$ state lying just below the Fermi level ($E_F$) at room temperature, strong hybridization occurs at low temperatures. The reconstructed SS due to the $c$–$f$ hybridization at 20 K is metallic and is continuously dispersed across the bulk bandgap of the KI between $E_F$ and $\sim$50 meV binding energy. This low-temperature SS behaviour agrees with the expected behaviour for SS on TKIs. Moreover, these results demonstrate the temperature-dependent holistic reconstruction of 2D states localized on the KI surface driven by the Kondo effect.

## Results

**Characterization of the $YbB_{12}$(001) clean surface.** Figure 1b shows the low-energy electron diffraction (LEED) pattern observed after the cleaning process (see Methods section). As shown by the sharp spots and low backgrounds, a well-ordered clean $YbB_{12}$(001) surface was obtained. In addition to the integer order spots corresponding to a (001) in-plane lattice constant

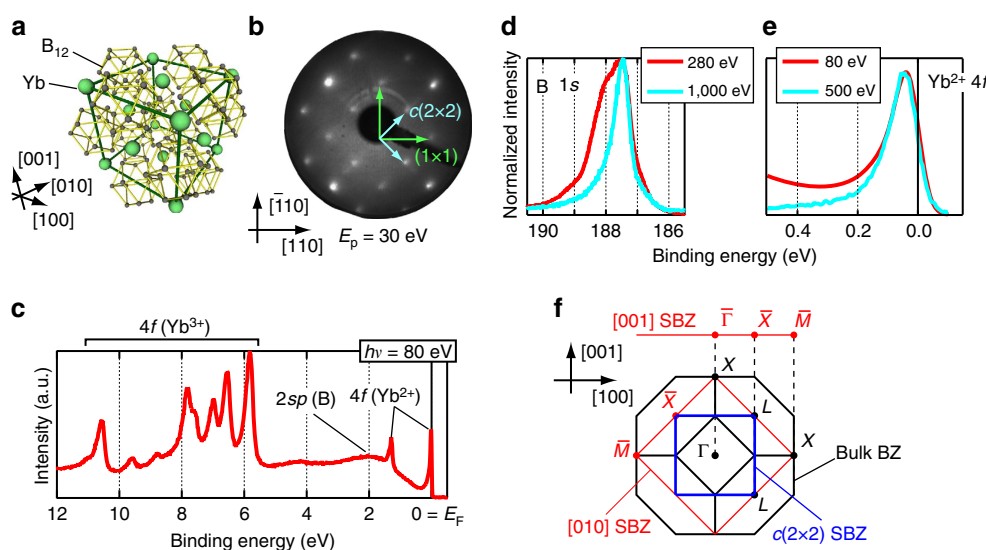

**Figure 1 | Characterization of the $YbB_{12}$(001) clean surface.** (**a**) Crystal structure of $YbB_{12}$ (ref. 28). (**b**) A LEED pattern of the clean $YbB_{12}$(001) surface at room temperature. Arrows indicate the surface unit vectors. (**c–e**) Angle-integrated photoelectron spectra taken at 20 K with photon energies at (**c**) 80, (**d**) 280/1,000 and (**e**) 80/500 eV. In (**e**) the spectrum taken at 80 is convolved with a Gaussian (full-width at half-maximum of 75 meV) for comparison with other spectra at 500 eV taken with lower energy resolution. (**f**) Schematic drawing of the three-dimensional Brillouin zone of the $YbB_{12}$ single crystal and its projection onto the (001) SBZ.

(5.28 Å), we found fractional order spots showing $c(2 \times 2)$ surface periodicity (Fig. 1b). The LEED pattern shows fourfold rotation symmetry, which is expected from the bulk crystal structure (Fig. 1a).

Figure 1c shows the wide-valence band spectrum for the $YbB_{12}(001)$–$c(2 \times 2)$ surface taken at $hv = 80$ eV. The $Yb^{2+}$ and $Yb^{3+}$-$4f$ levels are observed at similar energy positions to those measured with scraped or cleaved $YbB_{12}$ single crystals[21,22]. The major difference observed in this work is the absence of $Yb^{2+}$ components around 0.9 and 2.2 eV, which are assigned to be surface components. It would be because the Yb atoms at the topmost surface are desorbed during the heating process. Indeed, as shown in Fig. 1d, we found a clear surface component at the B-$1s$ level at the binding energy of 188 eV with a surface-sensitive condition ($hv = 280$ eV, kinetic energy $\sim 90$ eV), but this appeared as a weak tail in the bulk-sensitive measurement ($hv = 1,000$ eV). Regarding the $Yb^{2+}$-$4f$ levels, Fig. 1e shows almost the same peak positions with surface/bulk-sensitive conditions, supporting the above assumption that Yb atoms are not located at the topmost surface layer but in the deeper layers.

**SS on $YbB_{12}(001)$**. Figure 2 shows ARPES intensity plots taken below 20 K: Energy and momentum distribution curves (MCDs) are shown in Supplementary Figs 1–3. As shown in Fig. 2a, square constant energy contours (ECs) are observed at the binding energy of 200 meV. These contours fold with respect to the $c(2 \times 2)$ surface Brillouin zone (SBZ) boundary, suggesting they originate from the surface. Figure 2b shows the intensity plots of the MDCs along the dashed line in Fig. 2a as a function of probing photon energies. As indicated by the dashed line, the MDC peak corresponding to the square EC does not change its in-plane wave vector, indicating that this state is a 2D surface state.

Figure 2c,d are the band dispersions along [100] and [110], respectively. In what follows, we use the Miller indices based on the bulk crystal structure to define the direction in reciprocal space. The surface state ($S$ in Fig. 2c) identified above shows a steep dispersion below 100 meV. Dashed lines superposed on Fig. 2d are the calculated bulk bands based on the local density approximation (LDA) plus Gutzwiller method[20]. We captured the bulk bands around the $X$ point of bulk Brillouin zone. From the calculated bulk bands along $\Gamma$–$X$ (see ref. 20), the highly dispersive conduction band clearly has its minimum at $X$; this means the bulk band dispersion around $X$ is almost at the lower edge of the projected bulk bands in SBZ around $\bar{\Gamma}$ onto which the $\Gamma$–$X$ line is projected. Below 100 meV, the $S$ band dispersing is almost parallel to the calculated outer bulk band, which would be nearly parallel to the edge of the projected bulk bands. Note that $S$ shows no dispersion along $k_z$, which clearly differs from the bulk conduction band dispersing along the surface normal (see ref. 20). Such SS dispersion along the edge of the projected bulk bands are known for so-called Shockley-type SS on noble metal surfaces[23] and for surface resonances localized in the subsurface region of semiconductor surfaces[24]. This type of SS originates from bulk bands, but it is localized in the surface/subsurface region because of the truncation of the three-dimensional periodicity of the crystal. Some features of the ARPES plot, such as the inner bulk band below 100 meV and the number of $4f$ branches around $\bar{\Gamma}$ just below $E_F$, does not agree with the calculation. This disagreement is discussed in the following part.

In addition to the highly dispersive $S$ band, there is a less dispersive state around 30 meV ($F$ in Fig. 2c), probably originating from the $Yb^{2+}$-$4f$ bands. The $F$ band appears as single peak at $\bar{\Gamma}$ ($F_0$) but splits to two branches ($F_{up}$ and $F_{dn}$) with

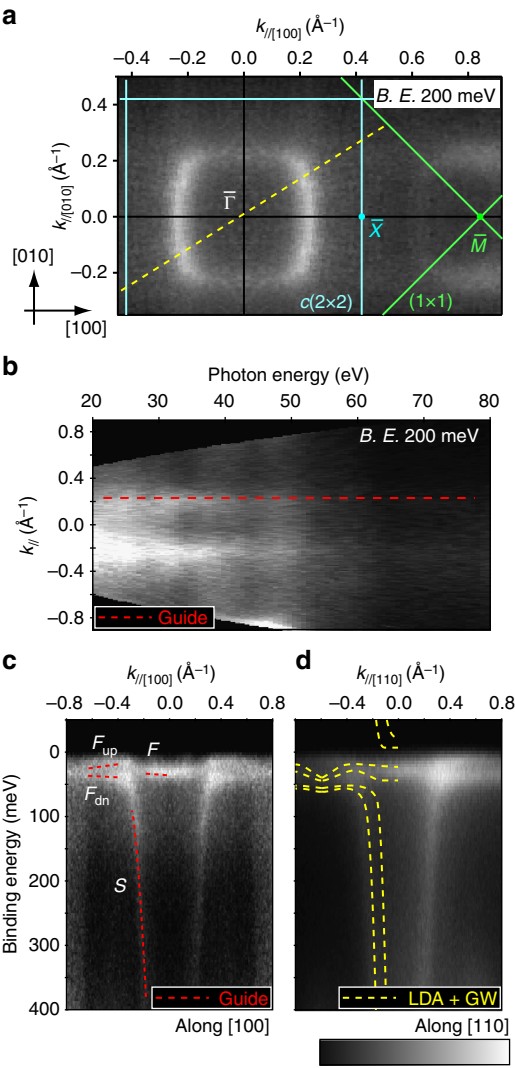

**Figure 2 | ARPES data taken below 20 K.** The intensity plots are shown in grayscale: white area represents high intensity. (**a**) Constant EC at the binding energy of $200 \pm 10$ meV ($hv = 50$ eV). Thin lines represent the SBZ boundary. Arrows indicate the Miller indices based on the bulk crystal structure. The indexes used in the following parts obey these definitions. (**b**) MCDs along a dashed line in (**a**) at the binding energy of $200 \pm 10$ meV as a function of photon energies. The dashed line provides a guide to the eye. (**c,d**) ARPES intensity maps along (**c**) [100] and (**d**) [110] measured with $hv = 53.5$ eV. Dashed lines in the left region of (**c**) indicate the observed bands named $S$, $F$, and $F_{dn/up}$. Those in (**d**) are calculated bulk bands around the $X$ point of bulk BZ captured from ref. 20.

the wave vector larger than $0.4 \text{ Å}^{-1}$ (smaller than $-0.4 \text{ Å}^{-1}$). These states cross with the $S$ band around $0.3 \text{ Å}^{-1}$ where they apparently hybridize with each other. Such behaviour suggests $c$–$f$ hybridization driven by the Kondo effect. The difference between the current case and other reported Kondo systems is that the hybridization occurs between SS, suggesting 2D $c$–$f$ hybridization. The nature of this $c$–$f$ hybridization as well as the origin of the $F$ band splitting dependence on $k$ is examined in more detail in the following section.

**Temperature-dependent reconstruction of SS.** To examine the origin of the hybridization between the surface branches, we measured the temperature-dependent ARPES images at

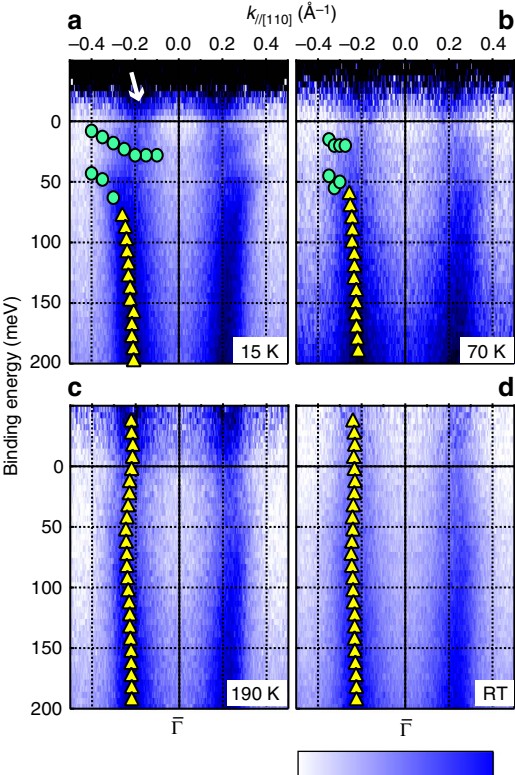

**Figure 3 | ARPES intensity plots near $E_F$ at different temperatures.**
(**a**–**d**) ARPES data are divided by the Fermi distribution function at each sample temperature convolved with the instrumental resolution. All data were taken along [110] with $hv = 16.5$ eV. The intensity plots are symmetrized with respect to $\bar{\Gamma}$ ($k = 0$ Å$^{-1}$) and shown in a colour scale: dark blue area represents high intensity. Circle (triangle) markers are the peak positions from ARPES momentum (energy) distribution curves (these curves are shown in Supplementary Fig. 2). The white arrow in (**a**) indicates a state lying at $E_F$.

$hv = 16.5$ eV as shown in Fig. 3. The ARPES spectra were divided by the Fermi distribution function convolved with the instrumental resolution to observe the unoccupied states. At 15 K, two separate bands are observed. Compared with the ARPES image at Fig. 2c, the lower branch is composed of a continuous connection between $S$ and $F_{dn}$, namely the $c$–$f$ hybridization band. The other branch lies at $\sim 30$ meV with slight dispersion. Apparently, the binding energy of this branch agrees with that of $F_{up}$. However, these 4$f$ states at $\bar{\Gamma}$ and $k_{//[110]} > 0.4$ Å$^{-1}$ are not visible because the photoemission cross-section of 4$f$ electrons is negligibly small at this photon energy. The 4$f$ states are observed as clear peaks at $k_{//[110]} = 0.2$–$0.3$ Å$^{-1}$ probably due to hybridization with the $S$ band, which should have Yb 5$d$ and/or B 2$sp$ character. As a result of this hybridization, the $F$ branches at 0.2–0.3 Å$^{-1}$, which are near the crossing point between $S$ and $F$, gain the orbital character of $S$ and become visible by ARPES at this photon energy.

At 70 K, the dispersion is almost the same as that at 15 K, but the separation between the 4$f$ branches is more diffuse than that at 15 K. At 190 K, $S$ becomes a continuous metallic band across $E_F$, and the 4$f$ states almost disappear. On $S$, there is an undulation of the SS dispersion, which is possibly a precursor of the hybridization with the 4$f$ states. At room temperature, the undulation of $S$ is smaller but still visible. The overall evolution of surface bands indicates that hybridization between the highly dispersive $S$ band and the almost-localized 4$f$ states depends on

the temperature, which is driven by the Kondo effect. This temperature-dependent behaviour, namely the disappearance of $c$–$f$ hybridization at 190 K, agrees well with the previous reports about bulk states; the Kondo temperature of bulk YbB$_{12}$ is $\sim 220$ K (ref. 22), where the $c$–$f$ gap size converges to zero. In addition, the remaining undulation of $S$ is also consistent with the mid-infrared peak that still survives at room temperature[18]. These results suggest a close relationship between the ARPES results and the bulk electronic states. However, Fig. 2b indicates the 2D nature of the $S$ band. Thus, the $F$ and $F_{up/dn}$ states intermixing with $S$ should also be localized in the surface/subsurface region. One possible interpretation of this case is that the nature of the $c$–$f$ hybridization on the YbB$_{12}$(001) surface is similar to that of the bulk bands.

**Topological origin of SS.** In addition to the states presented above, there is another state lying at $E_F$ as indicated by the white arrow in Fig. 3a. We focus on this state at low temperature in this section. Figure 4a,b shows the band dispersion along [100] taken at 16.5 and 53.5 eV, respectively. At both photon energies, a new state crossing $E_F$ at $k_{//[100]} \sim 0.18$ Å$^{-1}$ is clearly observed. As shown in Fig. 4d, this state is absent at room temperature and is replaced with another SS with dispersion that is almost parallel to the bulk $c$ band; this SS is the same as that observed in Fig. 3d. This state is degenerate with the 4$f$ ($F$) band at $\bar{\Gamma}$. Moreover, this 4$f$ band disperses between $E_F$ and 40 meV and becomes degenerate with the lower 4$f$ branch ($F_{dn}$) at $\sim 0.85$ Å$^{-1}$ ($\bar{M}$ of the (1 × 1) SBZ, see Fig. 4e). The complete SS dispersion exhibits a continuous connection across the Fermi level as well as the bulk $c$–$f$ hybridization gap, whose size is at most 100 meV[18], with two-fold degeneracy at high symmetry points of the (1 × 1) SBZ. This metallic surface state can explain the remnant conduction path of YbB$_{12}$ observed at low temperatures (Iga—private communication). Note that no states corresponding with the metallic state observed by ARPES were observed in the bulk bands[17,18].

The dispersion of the surface state observed here agrees with the expected behaviour for topological SS[1]: continuous dispersion across the bulk bandgap with Kramers degeneracy at the surface time-reversal invariant momentum. Indeed, the EC at $E_F$ taken with 16.5 eV photons indicates a closed Fermi contour surrounding $\bar{\Gamma}$, one of the surface time-reversal invariant momentum on (001), as shown in Fig. 4c. This closed EC surrounding $\bar{\Gamma}$ agrees with the expected behaviour for SS with non-trivial topological order defined with time-reversal symmetry.

One of the most prominent characteristics of topological SS is the helical spin polarization. In other words, the electrons in such SS are spin polarized towards the direction perpendicular to both the wave vector $k_{||}$ and the surface normal. To elucidate such a feature, we have measured the circular dichroism (CD) of ARPES as shown in Fig. 5a. CD of ARPES can show the orbital-angular-momentum (OAM) polarization of the electronic state[25]. As depicted in Fig. 5b, the incident circularly polarized photons were in the (010) plane in our experimental geometry, and hence the helicity of the photons should probe the OAM polarization along [001] or [100] which is the helical direction explained above in the measurement plane along $k_{||}$ [010]. Figure 5a clearly shows the CD contrast around $\bar{\Gamma}$ in the second SBZ. The contrast around 0.2 Å$^{-1}$ at $E_F$ agrees with the metallic SS observed at $k_{//[100]} \sim 0.18$ Å$^{-1}$ in Fig. 4a,b. The sign of the contrast inverts with respect to $\bar{\Gamma}$. Such behaviour agrees with what is expected for topological SS with helical spin polarization parallel/anti-parallel to [100]. Moreover, spin-resolved photoelectron spectra shown in Fig. 5d–g clearly indicates the spin polarization

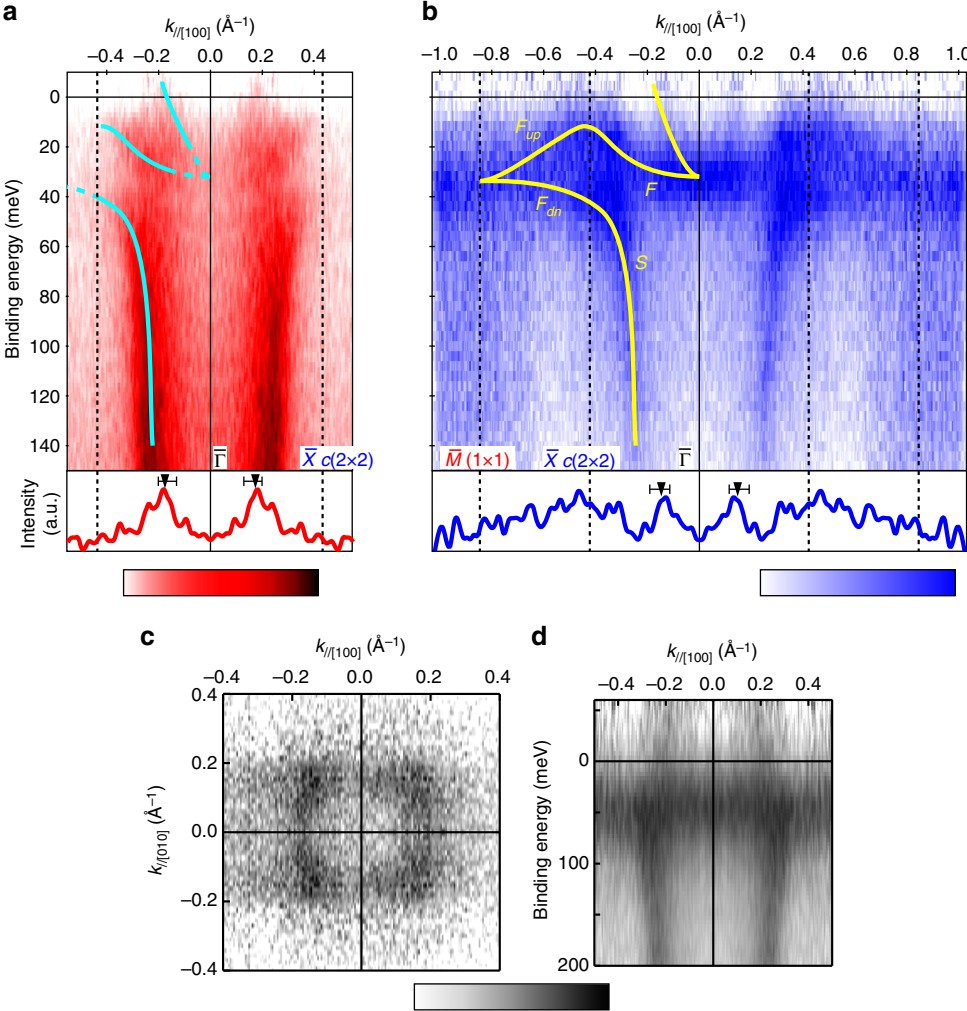

**Figure 4 | Topological surface-state dispersions observed by ARPES. (a,b)** ARPES intensity plots along [100] near $E_F$ taken with (**a**) 16.5 and (**b**) 53.5 eV photons and MCDs at $E_F$ ($\pm 10$ meV). The data is divided by the Fermi distribution function at the sample temperature (20 K for (**a**) and 14 K for (**b**)) convolved with the instrumental resolution. Lines are drawn as guides to the eye. (**c**) Constant EC around $E_F$ ($\pm 10$ meV) taken with 16.5 eV photons at 14 K. (**d**) ARPES image with 53.5 eV photons measured at room temperature. The ARPES intensity plots are symmetrized with respect to $\bar{\Gamma}$ ($k = 0$ Å$^{-1}$) and shown in a colour scale: dark (red, blue, and black for (**a**), (**b**), and (**c, d**) respectively) areas represent high intensity.

parallel/anti-parallel to [100] around the Fermi level. Although the OAM polarization does not always correspond one-to-one to the spin polarization[26], the spin-resolved spectra show that the CD contrast in Fig. 5a does correspond to the spin polarization of the metallic SS. These CD and spin-resolved ARPES results strongly suggest the helical spin polarization of the metallic SS around $\bar{\Gamma}$ on YbB$_{12}$(001) observed in this work, which is the evidence of topological SS.

The non-trivial topological order defined with time-reversal symmetry, as suggested above, does not agree with the theoretical prediction based on LDA + Gutzwiller method[20] which showed non-trivial mirror Chern number. It would be due to the accuracy limit of the calculation. Although the LDA + Gutzwiller method can calculate the correlated electronic states much better than conventional LDA, the agreement with the experiment is not perfect. The topological order of YbB$_{12}$ can change in very small energy scale, the order of 10 meV, between the Yb-4$f$ levels with different parities. Another cause of the disagreement is the surface atomic structure. As is explained with Fig. 1, the YbB$_{12}$(001) surface in this work would be terminated by B atoms without Yb and forms $c(2 \times 2)$ surface superstructure. In contrast, such surface atomic structure was not implemented to the DFT

calculation. Although the surface atomic structure does not affect the topological order, the dispersion of the SS can drastically change accordingly[27]. Therefore, it is not so strange that there is a disagreement between ARPES and DFT. Actually, as shown in Fig. 2d, some calculated bulk bands based on LDA + Gutzwiller method are not experimentally observed by ARPES. Following theoretical research about the discrepancy observed in this work might provide further insight into the surface electronic structure calculation with strong electron correlation.

## Discussion

In summary, we discovered the new surface state on a clean surface of the YbB$_{12}$(001) Kondo insulator and surveyed its temperature-dependent reconstruction using ARPES. The state is metallic and shows no dispersion along $k_z$, indicating its surface localization. While the SS does not hybridize with the Yb-4$f$ state lying just below the Fermi level ($E_F$) at room temperature, strong hybridization occurs at low temperatures. The reconstructed SS due to the $c$–$f$ hybridization at 20 K is metallic and dispersed continuously across the bulk bandgap of the Kondo insulator between $E_F$ and the binding energy of $\sim 50$ meV. This SS

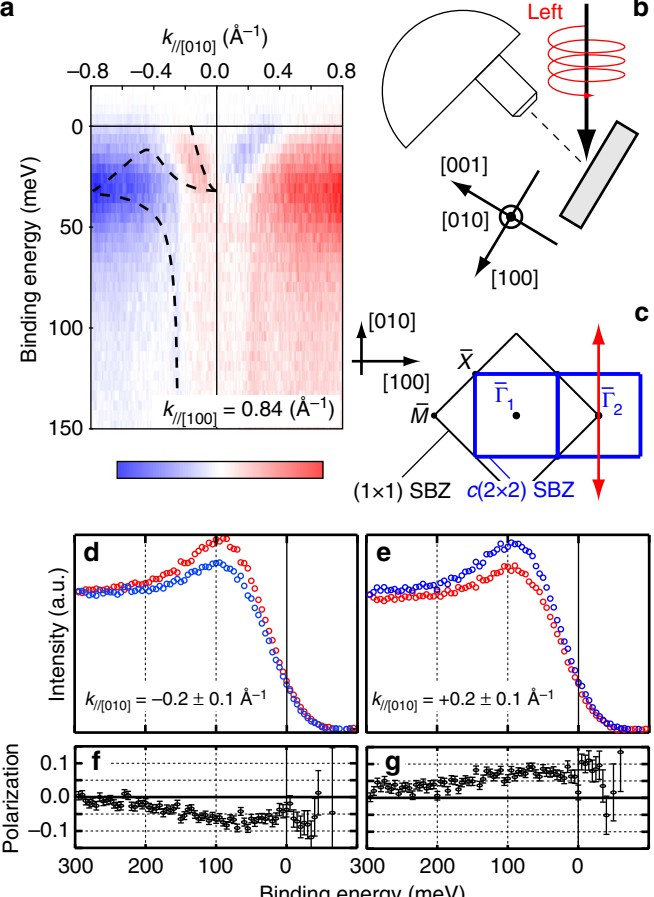

**Figure 5 | Orbital and spin polarization of SS.** (**a**) ARPES curcular-dichroism plot taken with $hv = 50$ eV photons at 10 K. Dichroism is obtained by subtracting the intensities of right-circularly polarized photons from those of left-circularly polarized ones. Red (blue) areas represent the maximum (minimum) dichroism values as shown by the colour scale. A dashed line in the left side is the same as that in Fig. 4b, a guide of the observed band dispersion from ARPES intensity plot. (**b**) Experimental geometry in this work for ARPES circular-dichroism plot shown in (**a**). (**c**) SBZ. An arrow indicates the probed region in (**a**). (**d,e**) Spin-resolved ARPES energy distribution curves (EDC) taken at 20 K. Blue (red) spectra corresponds with the spin polarization parallel (anti-parallel) to [100]. (**f,g**) Spin polarization of the EDC shown in (**d,e**) respectively. Positive (negative) value represents the spin polarization parallel (anti-parallel) to [100]. Errors are standard statistical errors from photoelectron counting.

dispersion and spin/OAM polarization at low temperatures agree with the expected behaviour for SS on TKIs. Moreover, these results demonstrate the temperature-dependent holistic reconstruction of 2D states localized on the surface of the Kondo insulator driven by Kondo effect. Further study about quantitative spin polarization analysis about $c$–$f$ hybridization would provide further insight into the topological order and electron correlation in the $YbB_{12}$ surface electronic structure.

## Methods

**ARPES and spin-resolved ARPES measurements.** The ARPES measurements were performed with synchrotron radiation at the CASSIOPÉE beamline of the SOLEIL synchrotron, the BL7U beamline of UVSOR-III and the BL-2A MUSASHI beamline of the Photon Factory. The photon energies used in these measurements ranged from 15 to 1,600 eV. The incident photon is linearly polarized and the electric field of the photons lies in the incident plane (so-called $p$ polarization) for the measurements shown in Figs 1–4 and circularly polarized for Fig. 5. The

photoelectron kinetic energy at $E_F$ and the overall energy resolution of each ARPES set-up ($\sim 20$ meV for ARPES and $\sim 80$ meV for wide-valence spectra shown in Fig. 1d,e) were calibrated using the Fermi edge of the photoelectron spectra from Ta foils attached to the sample. Spin-resolved ARPES measurements were performed at HiSOR BL9B with linearly polarized photons at 50 eV. The energy resolution was 80 meV and the effective Sherman function of the spin detector was set to 0.3. The acceptance angle for the spin detector was $\pm 1.5°$.

**Sample preparation.** Single crystalline $YbB_{12}$ was grown via the floating-zone method using an image furnace with four xenon lamps[17]. The crystal was cut with a diamond saw along the (001) plane based on an *in-situ* Laue pattern and was then polished in air until a mirror-like surface plane was obtained. The polished $YbB_{12}$(001) crystal was heated to 1650 K in ultra-high vacuum chambers for $\sim 10$ s. For heating, a SiC wafer was underlaid below $YbB_{12}$(001); heating the SiC wafer with direct current simultaneously heats the attached $YbB_{12}$.

**Data availability.** The data supporting the findings of this study are available within the article and its Supplementary Information, or from the authors upon request.

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

## Acknowledgements

We thank J. Kishi, Y. Takeno and Y. Negoro for their support during general experiments. For preliminary experiments to obtain a clean sample surface, we thank K. Imura, T. Hajiri and T. Ito for their support. We acknowledge D. Ragonnet and F. Deschamps for their support during the experiments on the CASSIOPÉE beamline at the SOLEIL synchrotron. We also thank S. Wu and T. Yoshikawa for their support for the spin-resolved ARPES measurements at HiSOR. Part of the ARPES experiments were performed under UVSOR Proposal Nos. 26-540 and 27-542 and Photon Factory Proposal No. 2015G540. This work was also supported by the JSPS Grant-in-Aid for Scientific Research Activity Start-up (Grant No. JP26887024), (B) (Grant No. JP15H03676) and the Murata Science Foundation.

## Author contributions

K.H. and Y.O. conducted the ARPES experiments with assistance from M.M., S.-i.I, K.T., J.R., P.L.F., F.B., A.T.-I., R.Y., M.K., K.H. and H.K. K.H. and Y.O. also performed the spin-resolved ARPES supported by K.S. and T.O. F.I. grew the single-crystal samples. H.M. conducted the preliminary experiments to obtain a well-ordered clean sample surface. Y.O. and S.-i.K. wrote the text and were responsible for the overall direction of the research project. All authors contributed to the scientific planning and discussions.

## Additional information

**Competing financial interests:** The authors declare no competing financial interests.

