## [Peer Review File · Nature Communications]

Reviewers' comments:

Reviewer #1 (Remarks to the Author):

In this manuscript, the authors use ARPES to examine the nature of the surface state of YbB12, trying to establish its topological nature. The data presented do provide some signature for a topological Kondo insulator, and it is indeed the first time that such high quality data have been taken on YbB12. This is an important and timely subject. However, before I could make a positive recommendation, I would like to see the following improvements.

1. The measured Fermi surface and surface states are quite different from that predicted in Ref.19, while this calculation is the main motivation of this work. For many topological phases, the agreement between the calculation and ARPES measurement was the main argument for the topological nature. Because the band calculations can be quite off for Kondo systems (even though LDA+Guzwiller has been implemented in Ref.19), this would make the findings here even more interesting. However, a direct comparison and explanation should be given in the discussion.
2. The authors mentioned that the spin and orbital angular momentum textures need to be measured in the future. This is a major caveat of this paper for arguing the topological nature, particularly considering the poor agreement with the band calculations. Spin-resolved work can be tricky and resource-limited, but circular dichroism ARPES should be easy to do, which would at least give hints on the chirality of the orbital angular momentum. Such data should be included, otherwise the conclusions on the topological nature is rather shaky.
3. There are still some problems with the data presentation, for example, color scale bars are missing in the figures. The data points (circles and triangles) in Fig.3 do not match the image plot, and it would be helpful if raw EDC and MDC for the data analysis are presented in the supplementary materials.

Reviewer #2 (Remarks to the Author):

Authors study the electronic structure of Kondo insulator YbB12 using angle resolved photoemission spectroscopy. YbB12 shows saturation of resistivity similar to SmB6, which might derive from topological surface states. Indeed, it has been predicted to be a crystalline topological insulator (Weng

PRL 112, 016403 (2014)). Thus direct observation of the topological surface states is of great interest to the community.

I am not convinced that the authors have observed these topological surface states. I believe that quality of the data is very poor to reach this conclusion. Both in Fig. 2 or Fig. 3, the data is quite scattered and particularly in Fig. 3, the bands which are "guide to the eye" push ones imagination. It is not clear to me why the ARPES lack the sufficient resolution in this case.

Moreover there is almost no connection with first principle calculations. Since YbB12 is predicted to be a topological crystalline insulator, it is expected to have 2 Dirac cones whereas as far as I can see, the authors only observe single Fermi surface around Γ point (Fig. 4. c). The paper lacks comparison with theory. As it stands, I do not suggest for publication in Nature Communications.

Reviewer #3 (Remarks to the Author):

Surface Kondo Effect and Non-trivial Metallic State of the Kondo Insulator YbB12

K. Hagiwara et al. present angle-resolved photoemission spectroscopy studies on topological Kondo insulator candidate YbB12. Authors found the metallic like surface state on the band structure of this compound. They showed temperature dependent reconstruction related with Kondo effects. Authors suggested the presence of topological surface states in YbB12. Establishing the TKI state in YbB12 will be interesting for the community working on Topological Kondo Insulator (TKI) phase because it has bigger gap (~ 40 meV) as compared with previous TKI candidate SmB6 (~ 15 meV).

However, looking at the ARPES data authors claim seems to be over-simplified. The ARPES data is very low quality and the dashed lines over the plots do not follow the real band structure for example in Fig 2b-d. The band-mapping plot in Fig. 2a looks good but it is well below the Fermi level (~ 200 meV). The temperature dependent data (even divided by Fermi function) do not show any obvious temperature dependent hybridization information.

Furthermore for the signature of the topological insulator phase, authors argue the robustness of the surface state independent of cleaving planes and reconstructions. However, spin-momentum locking behavior is a smoking gun to prove the topological order. Spin-resolved ARPES data or circular dichroism (CD) data is helpful. Current state of manuscript shows the poor data quality and insufficient experimental evidences to claim TKI. Even with the bigger Kondo gap situation here in YbB12, the Dirac like linearly dispersive band is absent in the data. Based on my opinion, this paper does not advance the understanding of topological Kondo insulator properties beyond what is already known in SmB6, I cannot recommend its publication in Nature Communications.

Minor suggestion:

1. This manuscript did not cite the early ARPES works on SmB6 by Hasan, and Hong Ding/PSI group. It would be better to give them credit for their earlier works.

REVIEWERS' COMMENTS:

Reviewer #1 (Remarks to the Author):

The authors have addressed my comments with circular dichroism and spin-resolved ARPES, which give strong support to their claims. The revised discussion is also satisfactory. Therefore I recommend it to be published in Nature. Commun.

I also wish to point out that the data quality of these hard-to-cleave systems are always not as good as those of 2D materials, however, the current data are already so far the best for this important material. Considering the experimental difficulty and novelty, I would recommend its publication even the data quality is not as nice as that of Bi₂Se₃, for example.

Reviewer #2 (Remarks to the Author):

Referee Report for Hagiwara et al. "Surface Kondo Effect and Non-Trivial Metallic State of the Kondo Insulator YbB₁₂"

Authors have substantially improved the quality of the data and addressed all my questions. They address possible reasons for disagreement with the theoretical calculations. I believe the results presented in this paper are of great interest to the community. I would suggest for publication in Nature Communications.

Reviewer #3 (Remarks to the Author):

I read the revised manuscript and referee reply from the authors. The manuscript is much improved now. Furthermore, the authors added spin and CD data in the maintext which support the possible topological origin of the metallic state. Even though there is no direct visualization of Dirac like dispersion and Dirac point in the data (which suggests that there is still a room to improve sample quality and get better cleaved surface), the T-dependent data, CD data, spin-resolved data and interpretation suggest the possible topological origin. This manuscript will stimulate further discussion in TKI system, therefore I support its publication in Nat. Commun.

(Replies to the reviewer #1)

We are pleased that the reviewer evaluates the impact, importance, and the quality of the experimental data presented. Thanks to the suggestions/requests from the reviewer, we could improve our manuscripts (MS) significantly.

(1) Disagreement between calculation and ARPES

The discussion about the disagreement between the calculation (ref. 21 in the revised MS: Weng *et al.*, PRL112) and our ARPES results is added in p. 6 and 8-9 in the revised manuscript. In short, we proposed two possible cause of the disagreement. First, the calculation can be off for Kondo systems, as the reviewer suggested in the previous report. And second, the calculation did not take the surface atomic structure ($c(2 \times 2)$ periodicity and absence of Yb) into account.

(2) Spin-resolved or circular dichroism ARPES measurements

We added the circular dichroism (CD) of ARPES and spin-resolved ARPES as Figure 5 in the revised MS. The CD contrast was clearly observed for the metallic surface state which we assumed to be a topological surface state. The sign of the CD contrast changed with respect to Γ in 2nd surface Brillouin zone (SBZ), suggesting a helical orbital-angular-momentum polarization which is expected for topological surface states. The spin-resolved data also showed the spin polarization of the surface state at the Fermi level.

Concerning this point, we removed the discussion about the absence of the surface states around Γ in 2nd SBZ because it is not correct. The surface state was indeed observed as shown in Fig. 5. The absence in Fig. 4 (b) would be due to the selection rule of photoelectron excitation process. Actually, Fig. 2 (a) shows that the surface states is evident along [100] in the 1st SBZ but almost vanishes in 2nd SBZ. Instead, the surface state become intense along [010], where the CD in Fig. 5 was probed.

(3) Problems with the data presentation

We added colour scales in Figures 2, 3, 4 and 5. In addition, we have also made a supplementary material showing the ARPES energy and momentum distribution curves.

With these revisions, we hope the revised MS could meet the requests from the reviewer.

(Replies to the reviewer #2)

We are pleased that the reviewer evaluates the impact and importance of this work. After the revision of the manuscript (MS) in order to meet the criticisms from the reviewer, we could improve it significantly.

(1) Quality of the data (we numbered the comments from the reviewer for better readability)

Thank you for the comment. For better presentation, we made a supplementary material showing the ARPES energy and momentum distribution curves corresponding to the ARPES intensity plots in Figures 2, 3 and 4. From these spectra, one can find the peak positions without ambiguity.

In addition, we have also added an ARPES circular dichroism plot and spin-resolved ARPES spectra (Figure 5). The sign of the CD contrast changed with respect to Γ in 2nd surface Brillouin zone (SBZ), suggesting a helical orbital-angular-momentum polarization which is expected for topological surface states. The spin-resolved spectra shows the spin polarization of the surface state at the Fermi level, indicating that the CD contrast in this work has one-to-one correspondence to the spin polarization. These data suggest the origin of the metallic surface states from the topological order defined with time-reversal symmetry.

(2) Disagreement between calculation and ARPES

The discussion about the disagreement between the calculation (ref. 21 in the revised MS: Weng *et al.*, PRL112) and our ARPES results were added in p. 6 and 8-9 in the revised manuscript. In short, we proposed two possible cause of the disagreement. First, the calculation can be off for Kondo systems, as the reviewer suggested in the previous report. And second, the calculation did not take the surface atomic structure ($c(2 \times 2)$ periodicity and absence of Yb) into account.

With these revisions, we hope the revised MS could answer the criticisms from the reviewer.

(Replies to the reviewer #3)

We are pleased that the reviewer evaluates the impact and importance of this work. After the revision of the manuscript (MS) in order to meet the criticisms from the reviewer, we could improve it significantly.

(1) Quality of the data (we numbered the comments from the reviewer for better readability)

Thank you for the comment. For better presentation, we made a supplementary material showing the ARPES energy and momentum distribution curves corresponding to the ARPES intensity plots in Figures 2, 3 and 4. From these spectra, one can find the peak positions without ambiguity. These spectra also shows the change of the spectral shape depending on the temperature, as is discussed in p. 6 and 7.

For Fig. 2(d), the disagreement would be due to the imperfection of the theoretical calculation. A discussion about this point is added in p. 6 and 8 in the revised manuscript. In short, we proposed two possible cause of the disagreement. First, the calculation can be off for Kondo systems, as the reviewer suggested in the previous report. And second, the calculation did not take the surface atomic structure ($c(2 \times 2)$ periodicity and absence of Yb) into account.

(2) Spin-resolved or circular dichroism ARPES measurements

We added the circular dichroism (CD) of ARPES and spin-resolved ARPES as Figure 5 in the revised MS. The CD contrast was clearly observed for the metallic surface state which we assumed to be a topological surface state. The sign of the CD contrast changed with respect to Γ in 2nd surface Brillouin zone (SBZ), suggesting a helical orbital-angular-momentum polarization which is expected for topological surface states. The spin-resolved data also showed the spin polarization of the surface state at the Fermi level.

Concerning this point, we removed the discussion about the absence of the surface states around Γ in 2nd SBZ because it is not correct. The surface state was indeed observed as shown in Fig. 5. The absence in Fig. 4 (b) would be due to the selection rule of photoelectron excitation process. Actually, Fig. 2 (a) shows that the surface states is evident along [100] in the 1st SBZ but almost vanishes in 2nd SBZ. Instead, the surface state become intense along [010], where the CD in Fig. 5 was probed.

(3) Minor suggestion

Thank you for the kind introduction of early important works by Hasan and Hon Ding/PSI group. We added them to the reference list.

With these revisions, we hope the revised MS could answer the criticisms from the reviewer.

We would like to express our deepest acknowledge to the reviewers' comments and their voluntary efforts for our manuscript. We are pleased that all the reviewers recommended publication of our manuscript in Nature Communications without questions/criticisms, as shown in the following.

REVIEWERS' COMMENTS:

Reviewer #1 (Remarks to the Author):

The authors have addressed my comments with circular dichroism and spin-resolved ARPES, which give strong support to their claims. The revised discussion is also satisfactory. Therefore I recommend it to be published in Nature. Commun.

I also wish to point out that the data quality of these hard-to-cleave systems are always not as good as those of 2D materials, however, the current data are already so far the best for this important material. Considering the experimental difficulty and novelty, I would recommend its publication even the data quality is not as nice as that of Bi₂Se₃, for example.

Reviewer #2 (Remarks to the Author):

Referee Report for Hagiwara et al. "Surface Kondo Effect and Non-Trivial Metallic State of the Kondo Insulator YbB₁₂"

Authors have substantially improved the quality of the data and addressed all my questions. They address possible reasons for disagreement with the theoretical calculations. I believe the results presented in this paper are of great interest to the community. I would suggest for publication in Nature Communications.

Reviewer #3 (Remarks to the Author):

I read the revised manuscript and referee reply from the authors. The manuscript is much improved now. Furthermore, the authors added spin and CD data in the maintext which support the possible topological origin of the metallic state. Even though there is no direct visualization of Dirac like dispersion and Dirac point in the data (which suggests that there is still a room to improve sample quality and get better cleaved surface), the T-dependent data, CD data, spin-resolved data and interpretation suggest the possible topological origin. This manuscript will stimulate further discussion in TKI system, therefore I support its publication in Nat. Commun.